# Contextual Set Selection Under Human Feedback With Model Misspecification

**Shuo Yang** [1]   **Rajat Sen** [2]   **Sujay Sanghavi** [3]

## Abstract

A common and efficient way to elicit human feedback is to present users with a set of options, and record their relative preferences on the presented options. The contextual combinatorial bandits problem captures this setting algorithmically; however, it implicitly assumes an underlying consistent reward model for the options. The setting of human feedback (which e.g. may use different reviewers for different samples) means that there may not be any such model – it is *misspecified*.

We first derive a lower-bound for our setting, and then show that model misspecification can lead to catastrophic failure of the $C^2$UCB algorithm (which is otherwise near-optimal when there is no misspecification). We then propose two algorithms: the first algorithm ($MC^2$UCB) requires knowledge of the level of misspecification $\epsilon$ (i.e., the absolute deviation from the closest well-specified model). The second algorithm is a general framework that extends to unknown $\epsilon$. Our theoretical analysis shows that both algorithms achieve near-optimal regret. Further empirical evaluations, conducted both in a synthetic environment and a real-world application of movie recommendations, demonstrate the adaptability of our algorithm to various degrees of misspecification. This highlights the algorithm's ability to effectively learn from human feedback, even with model misspecification.

## 1. Introduction

In many practical applications, the objective is to learn the optimal set selection under a certain context. Take, for instance, the field of advertising, where multiple slots on a webpage are available for displaying ads. Here, the business owner aims to select the most effective set of ads, i.e., the ones that most engage the customer and potentially generate the most profit. Yet, in real-world datasets, *realizability* – the assumption of an exact functional relationship between human feedback and the features related to the context – is rarely satisfied. Misspecification happens in several ways: for example in personalized recommendation systems, multiple users might share an account occasionally. There can also be less benign causes such as non-genuine click traffic or reviews injected into the feedback by adversarial third-party agents. In this paper, we delve into the challenge of learning optimal set selection from online human feedback, even when faced with model misspecification.

The stochastic contextual bandit is a general framework for online learning problems, where at each round the learner observes a context $c \in \mathcal{C}$, then chooses one out of $m$ arms and observes a reward for only the arm chosen. The problem has been widely studied for the last two decades and has found uses ranging from online advertising, recommendations (Yue and Guestrin, 2011; Li et al., 2016) to drug testing and medical trials (Durand et al., 2018; Villar et al., 2015). The task is to minimize *regret* commonly defined as the difference in collected reward from an optimal policy. For instance in online advertising, the context can be the browser history and search query from a user, the arm can be a specific advertisement out of many possible choices and the reward can be a click or a purchase. The corresponding task is then to maximize the expected click or purchase rate. The main challenge is to achieve a delicate balance between *exploitation* i.e., choosing the best arm based on the current belief and *exploration* i.e., choosing a rarely chosen arm to gather more information about the environment.

Further, many real-world problems have many slots instead of one (e.g., choosing multiple ads to show). It is commonly referred to as the contextual combinatorial bandit problem, where at each time-slot the learner is allowed to select a set of $k < m$ arms. The problem has mostly been studied under the semi-bandit feedback where the learner can see the rewards from the individual arms in the chosen set. Prior works (Qin et al., 2014; Yue and Guestrin, 2011) have commonly made some reasonable assumptions on the value of a set of arms like sub-modularity or other smoothness assumptions. Both these works study the problem under a linear function class, i.e., the mean reward of an arm is an

[1]Department of Computer Science, University of Texas at Austin, TX, US [2]Google, CA, US [3]Department of Electrical and Computer Engineering, University of Texas at Austin, TX, US. Correspondence to: Shuo Yang <yangshuo_ut@utexas.edu>.

Interactive Learning with Implicit Human Feedback Workshop at ICML 2023.

unknown linear function of the arm's feature.

Model misspecification, as previously exemplified, is overwhelming in real-world problems with human feedback. This is especially true when the chosen function class is linear, which is a common choice for theoretically analyzed algorithms in contextual combinatorial bandits (Qin et al., 2014; Yue and Guestrin, 2011). Therefore, it is important to develop and analyze algorithms that can provably work under model misspecification without complete knowledge about the level of misspecification.

In this work we highlight the need for such robust algorithms in two ways, (i) theoretically, we show that a popular algorithm for contextual combinatorial bandit (Qin et al., 2014) can have a catastrophic failure in the presence of misspecification, while our algorithm can avoid it without the knowledge of the exact level of misspecification and (ii) in a practical movie recommendation task we demonstrate that our algorithms well adapt to the misspecification that naturally arises in the real-world dataset, and can outperform the commonly adopted algorithm.

The **main contributions** of this paper are as follows,

- **(Hardness)** We first characterize the hardness of model misspecification in linear contextual combinatorial bandits by providing a regret lower bound of $\Omega\left(k\epsilon\sqrt{(d-1)/(8\log(m))T}\right)$ as Proposition 4.1, where $m$ is the number of arms, $\epsilon$ the level of misspecification, $d$ the dimension of the arms' features and $T$ the time-horizon. We also show that a popular algorithm for this setting $C^2UCB$ (Qin et al., 2014) can have a catastrophic failure in the presence of model misspecification, leading to arbitrarily large regret on some problem instances. In particular, the $C^2UCB$ has regret $kT/2$, which is significantly worse than the lower bound $O(k\epsilon T)$ when $\epsilon$ is small.

- **(Algorithms)** We first propose Algorithm 1 for the misspecified setting where the misspecification level $\epsilon$ is known to the learner. In Theorem 5.1, we show that the algorithm has a regret of $\widetilde{O}\left(d\sqrt{kT} + k\epsilon\sqrt{d}T\right)$, thus essentially matching our lower bound. Next we propose Algorithm 2 that works even when $\epsilon$ is unknown to the learner. Algorithm 2 corrals multiple instances of Algorithm 1 with different values of hypothetical $\epsilon$'s, and is shown to have near-optimal regret.

- **(Empirical Validation)** Through simulated experiments, we show that our algorithms can significantly outperform the $C^2UCB$ (i.e., ours achieve sublinear regret while $C^2UCB$ yields linear regret). We then validate our algorithm on the Movielens dataset (Harper and Konstan, 2015) in a setting similar to (Qin et al.,

2014). We show that our algorithm has slightly better performance than $C^2UCB$ in the original setting while outperforming $C^2UCB$ by a large extent in the presence of small model misspecification. Thus our algorithm can lead to robustness by preventing catastrophic failure, with little/no performance cost.

## 2. Related Work

The combinatorial multi-armed bandit problem has been studied extensively in the last decade. There are several works in the semi-bandit feedback setting (Chen et al., 2013; Combes et al., 2015; Kveton et al., 2015; Merlis and Mannor, 2019; Yang et al., 2021), where the learner observes the individual rewards of the arms in the set of arms chosen at each round, along with the total reward for the composite action. This is also the feedback model in our paper. The problem has also been studied under full-bandit feedback where only the composite reward of the whole set of arms chosen can be observed (Cesa-Bianchi and Lugosi, 2012; Lin et al., 2014; Agarwal and Aggarwal, 2018; Rejwan and Mansour, 2020).

The contextual version of the problem has been studied in (Yue and Guestrin, 2011) under the linear function class, where the value utility from choosing a set of arms is submodular in terms of the individual rewards. Qin et al. (2014) also study the problem under linear rewards, where the reward derived from a set of arms is a function of the individual rewards that satisfies some monotonicity and Lipschitz properties. We closely follow the model in (Qin et al., 2014), however as we will show both theoretically and empirically, their algorithm is not robust to misspecification and can undergo catastrophic failure. Our algorithm, on the other hand, can adapt to unknown model misspecifications. Note that the contextual combinatorial bandit problem has been studied under general function classes recently in (Sen et al., 2021), however, they can only adapt to known misspecification and their algorithm only provably works for the case when the set function is just the sum of individual rewards. In fact, we show that as a corollary of our results, we can extend their algorithm to the unknown misspecification case.

The linear bandits problem was first studied in the misspecified setting in (Ghosh et al., 2017), where they propose a robust algorithm that switches between OFUL (Abbasi-Yadkori et al., 2011) and UCB (Auer, 2002) based on a hypothesis test. It was shown in (Lattimore et al., 2020) that the OFUL algorithm itself can be easily made robust to known model misspecification. In fact our algorithm for the known misspecification case can be seen as an extension of the algorithm in (Lattimore et al., 2020) to the combinatorial setting. Takemura et al. (2021) have recently come up with a robust version of LinUCB (Chu et al., 2011) that works without knowledge of the misspecification level,

however their algorithm is not easily implementable. Foster et al. (2020) have recently shown how to adapt to model misspecification for contextual bandits under general function classes, however it is unclear how their algorithms can be adapted to the combinatorial setting. There is also a growing body of literature on adversarial corruption. Most works in this area like (Seldin and Slivkins, 2014; Lykouris et al., 2018; Gupta et al., 2019) study the non-contextual K-armed setting. A recent work (Bogunovic et al., 2021) has examined the problem in the context of Gaussian process bandit optimization. None of the prior works consider the combinatorial contextual bandits setting. The contextual bandit work in (Bogunovic et al., 2021) can tolerate epsilon up to $O(1/\sqrt{T})$. Our regret bounds degrade gracefully with any $\epsilon$, and we can achieve the same regret scaling with $T$. However, it is important to note that we can only handle an oblivious adversary which we believe is good enough for many real-world applications.

Note that all the above works in misspecified bandits try to achieve a regret scaling of $O(\epsilon T)$, which is also the natural lower bound for most of the problem settings. The desideratum is for this linear (in $T$) term to degrade gracefully with the level of misspecification. Indeed algorithms that are not robust to misspecification can have regret $O(cT)$ under $\epsilon$-misspecification where $c$ is a constant with $c \gg \epsilon$ (Lattimore et al., 2020). This is in particular concerning when $\epsilon$ is small (e.g., $\epsilon = T^{-1/3}$, as the algorithms have regret linear in $T$ while it is possible to achieve $O(T^{2/3})$ regret).

Closely related is a growing body of literature on model-selection via corralling different bandit algorithms both in the adversarial setting (Agarwal et al., 2017) and in the stochastic setting (Cutkosky et al., 2020; Arora et al., 2021). Our algorithm that can adapt to unknown misspecification is based on the stochastic corralling idea but explicitly adapted to model misspecification, which can lead to sharper guarantees and much simpler algorithms.

# 3. Problem Setting

We consider the following contextual combinatorial bandits problem: At every time step $t$, the online learner observes a context $c_t$ generated by the environment according to a fixed context distribution $\mathcal{D}_C$. For the context $c_t$, the arms in set $A = [m]$ are featurized as $\{\mathbf{x}_t(1), \cdots, \mathbf{x}_t(m)\} \subset \mathbb{R}^d$. Based on $\{\mathbf{x}_t(i)\}_{i \in [m]}$, the online learner selects $k$ distinct arms, which we denote as $S_t$.

After the set $S_t$ is played, the online learner observes a score $\widetilde{r}_t(i)$ for each arm $i$ in set $S_t$, and a reward $\widetilde{R}_t$ for the set $S_t$, both generated by the stochastic environment. Given the context $c_t$, the reward expectation of $S_t$ is denoted by $R_t := \mathbb{E}\left[\widetilde{R}_t | c_t, S_t\right]$. Correspondingly, the regret at time step $t$ is defined as $Reg_t = R_t^* - R_t$, where $R_t^* :=$

$\max_{S_t} \mathbb{E}[R_t | c_t, S_t]$, i.e., the expected reward of the optimal set given context $c_t$. The goal is to minimize the cumulative regret $R(T) = \sum_{t=1}^{T} Reg_t$.

In this paper, we focus on a setting where an arm's expected score $r_t(i) := \mathbb{E}\left[\widetilde{r}_t(i) | c_t, S_t\right]$ comes from a linear model with some misspecification (Section 3.1) and the expected reward of a set $S_t$ is determined by a function of $\{r_i\}_{i \in S_t}$, with some general properties (Section 3.2).

## 3.1. Misspecified Linear Model

The linear contextual combinatorial bandits problem has been widely studied under the well-specified linear model: $r_t(i) = \theta_*^\top \mathbf{x}_t(i)$ (Qin et al., 2014). Since practical datasets are unlikely to be well-specified w.r.t the linear function class, recent work has begun to study the misspecified linear model for contextual bandits (Lattimore et al., 2020).

Here we extend the line of recent work on the misspecified model to contextual combinatorial bandits. Specifically, we consider the case where the expected score $r_t(i)$ follows a misspecified linear model:

$$r_t(i) = \theta_*^\top \mathbf{x}_t(i) + \Delta_t^{(i)},$$

where $\Delta_t^{(i)}$ captures the misspecification, which can depend on the arm $i$, the played set $S_t$, and can potentially be different in different time step $t$. In this paper, we consider bounded misspecification, which can be formalized as the following assumption. Note that when it is clear from the context we will drop the superscript in $\Delta_t^{(i)}$ and just use the notation $\Delta_t$.

**Assumption 3.1** (Bounded Misspecification). For any time step $t \in [T]$, arm $i \in [m]$, and set $S$, we assume the misspecification to be uniformly bounded as $\left|\Delta_t^{(i)}\right| \leq \epsilon$.

The online learner might not know the misspecification level $\epsilon$. In subsequent sections, we first present a near-optimal algorithm for the known $\epsilon$ scenario. Then we propose an algorithm that can adapt to an unknown misspecification level. We show that our algorithms enjoy near-optimal regret guarantees.

## 3.2. Set Reward Function

The expected reward $R_t$ of set $S_t$ is a general function of the expected scores $r_t(i)$ for the arms $i \in S_t$ and the arms' embedding $\mathbf{x}$ therein. We denote the set reward function to be $f_\mathbf{x}(\mathbf{r}_t, S_t)$, where $\mathbf{x}$ is the embedding of the arms, $\mathbf{r}_t$ is the vector of all $m$ arms' expected scores: $\mathbf{r}_t := [r_t(1), \cdots, r_t(m)]$. For brevity, we use $f(\mathbf{r}_t, S_t)$ in the rest of our paper.

In order to develop efficient algorithms, we need to impose the following regularity assumptions (Qin et al., 2014):

**Assumption 3.2** (Monotonicity of $f$). For any $S$, if $\mathbf{r}(i) \leq \mathbf{r}'(i), \forall i \in S$, we have $f(\mathbf{r}, S) \leq f(\mathbf{r}', S)$.

**Assumption 3.3** (Lipschitz continuity of $f$). There exists a universal constant $C > 0$ such that for any two scores $\mathbf{r}, \mathbf{r}'$ and any $S$, we have $|f(\mathbf{r}, S) - f(\mathbf{r}', S)| \leq C\sqrt{\sum_{i \in S}[\mathbf{r}(i) - \mathbf{r}'(i)]^2}$.

Here we present two commonly seen examples that satisfy the assumptions above.

**Example 1** (Summation). $f(\mathbf{r}, S) = \sum_{i \in S} \mathbf{r}(i)$ clearly satisfies the assumptions.

**Example 2** (Probabilistic Coverage). *Consider the observed score $\widetilde{r}_t(i)$ being a Bernoulli random variable with expectation $r_t(i)$ and set reward function being $f(\mathbf{r}, S) = 1 - \Pi_{i \in S}(1 - \mathbf{r}(i))$. This $f(\mathbf{r}, S)$ measures the probability that the set $S$ has at least 1 positive scored arm. It is easy to verify that $f(\mathbf{r}, S)$ is monotone and 1-Lipschitz continuous w.r.t $\mathbf{r}$.*

The online learner may not have direct knowledge of the set reward function $f(\cdot, \cdot)$. Instead, we assume that the online learner has access to an oracle $\mathcal{O}$, which takes an estimate of the arm scores $\widehat{\mathbf{r}}_t = [\widehat{r}_t(1), \cdots, \widehat{r}_t(m)]$ as input and returns the corresponding optimal set: $\mathcal{O}(\widehat{\mathbf{r}}_t) = \arg\max_S f(\widehat{\mathbf{r}}_t, S)$.

Notice that there is no necessity for estimating the set reward function $f(\cdot, \cdot)$, as the online learner has the oracle $\mathcal{O}$. The main challenge in achieving optimal regret is to estimate the unknown parameter $\boldsymbol{\theta}_*$ in the presence of model misspecification $\Delta_t$.

# 4. The Hardness of Model Misspecification

Here we show the hardness of the linear contextual combinatorial bandits problem with model misspecification. We focus on the simple setting with set reward function $f(\mathbf{r}, S) = \sum_{i \in S} \mathbf{r}(i)$. The hardness is two-folds: *Information-theoretically,* we show a regret lower bound in the presence of model misspecification (Proposition 4.1). This lower bound holds for all possible algorithms and represents the hardness of model misspecification. *Algorithmically,* we show that a previously established near-optimal algorithm for contextual combinatorial bandits (i.e., not adapted to misspecification) can be arbitrarily far from the information-theoretic lower bound (Proposition 4.3). This corresponds to the hardness and challenge in designing effective algorithms.

## 4.1. Regret Lower Bound

The lower bound here is a natural extension of the lower bound for misspecified linear bandits (Lattimore et al., 2020). Recall that $m$ is the number of arms, $d$ is the dimension of arm's feature $\mathbf{x}_t(i)$. We have the following lower bound which holds both when the online learner knows or does not know the misspecification level $\epsilon$.

**Proposition 4.1** (Lower Bound for Misspecified Contextual Combinatorial Bandits). *There exists a set of arms' features $\{\mathbf{x}_t(1), \cdots, \mathbf{x}_t(m)\}, t \in [T]$, such that for any algorithm, there is a $\boldsymbol{\theta}_*$ for which*

$$R(T) \geq k\epsilon\sqrt{\frac{d-1}{8\log(m)}}\min\left(T, \frac{m}{2k}\right).$$

*Remark* 4.2. The interesting regime is when $m \gg kT$, i.e., there is a large number of arms and the online learner cannot play them all. The lower bound then becomes $R(T) \geq k\epsilon\sqrt{\frac{d-1}{8\log(m)}}T$. The linear in $T$ term is unfortunately unavoidable.

**Proof Sketch:** The proof follows by constructing a set of $m$ arms, with features almost orthogonal to each other. The misspecification is set such that all but the optimal arm has reward 0. Therefore any algorithm learns nothing until it includes the optimal arm into the played set, which in expectation takes $\min(T, \frac{m}{2k})$ steps. The environment can be constructed such that playing each sub-optimal set induces $k\epsilon\sqrt{\frac{d-1}{8\log(m)}}$ regret (details deferred to Appendix A). Combining the two parts completes the proof.

## 4.2. Catastrophic Failure from Not Adapting to Misspecification

Algorithms not adapting to misspecification can have an arbitrarily large gap to the lower bound in Proposition 4.1. Here we focus on contextual combinatorial bandits algorithm $C^2UCB$ (Qin et al., 2014, Algorithm 1), which is near-optimal for linear models but does not adapt to misspecification. Specifically, $C^2UCB$ estimates $\boldsymbol{\theta}_*$ by an online ridge regression based on the arms' scores.

$$\widehat{\boldsymbol{\theta}}_t = \left(\lambda\mathbf{I} + \sum_{\tau=1}^{t-1}\sum_{i \in S_\tau}\mathbf{x}_\tau(i)\mathbf{x}_\tau(i)^\top\right)^{-1}\left(\sum_{\tau=1}^{t-1}\sum_{i \in S_\tau}\widetilde{r}_\tau(i)\mathbf{x}_\tau(i)\right).$$

where $\lambda > k$ is the ridge regression regularization. The UCB for each arm $i$ is calculated as

$$\widehat{r}_t(i) = \widehat{\boldsymbol{\theta}}_t^\top\mathbf{x}_t(i) + \beta_t\sqrt{\mathbf{x}_t(i)\mathbf{V}_{t-1}^{-1}\mathbf{x}_t(i)}, \qquad (1)$$

where $\mathbf{V}_{t-1} = \lambda\mathbf{I} + \sum_{\tau=1}^{t-1}\sum_{i \in S_\tau}\mathbf{x}_\tau(i)\mathbf{x}_\tau(i)^\top$ and $\beta_t$ scales the uncertainty (variance) to give the right confidence interval. The set $S_t$ is constructed by taking the arms with top-$k$ $\widehat{r}(i)$. The next result states that $C^2UCB$ can be arbitrarily far from the regret lower bound in Proposition 4.1.

**Proposition 4.3** (Failure of Ignoring Misspecification). *For any $\epsilon > 0$, there exists a problem instance with horizon $T$, dimension $d = 2$ and misspecification level $\epsilon$, such that the*

*regret of $C^2UCB$ is $R(T) = kT/2$, which does not become smaller when $\epsilon$ is small.*

**Remark** 4.4. Comparing the regret in Proposition 4.3 with the corresponding regret lower bound of $\widetilde{\Omega}(k\epsilon T)$. We see that the regret of $C^2UCB$ is worse by a factor of $1/\epsilon$. This is clearly sub-optimal when $\epsilon$ is small.

**Proof Intuition.** By carefully setting the misspecification $\Delta_t$, a very small $\epsilon$ can lead to subtle differences between the estimate $\widehat{\theta}_t$ and the true parameter $\theta_*$, which in turn leads to the selection of a set $S_t$ having constant regret (i.e., not depending on $\epsilon$). The ratio of the actual regret $R(T)$ and the lower bound $k\epsilon T$ can therefore be larger than any constant $\xi$, by setting $\epsilon \approx \frac{1}{\xi}$.

**Remark** 4.5. As a comparison, we will present an algorithm in the next section (Algorithm 1) adapted to model misspecification. The algorithm is nearly optimal, in the sense that there exists a problem independent constant $\xi_0 \in \mathbb{R}^+$, such that $\frac{R(T)}{k\epsilon T\sqrt{\log T}} \leq \xi_0$ for all $k, \epsilon$ and $T$.

One particular regime of interest is when $\epsilon = \Theta(\frac{1}{\sqrt{T}})$. While the lower bound Proposition 4.1 suggests that it is possible to achieve $O(k\sqrt{T})$ regret, our lower bound problem instance shows that the $C^2UCB$ can only achieve a linear $\Theta(kT)$ regret. As we will see in the next section, our proposed algorithm (Algorithms 1 and 2) can achieve $O(k\sqrt{T})$ regret, which matches the lower bound and avoids the linear regret.

# 5. Known Misspecification Level $\epsilon$

In this section, we present the algorithm when the misspecification level $\epsilon$ is known. The algorithm incorporates the misspecification into the optimistic arm estimate, which gives a valid UCB for each arm in the presence of model misspecification. Further, we prove that the proposed algorithm is nearly optimal for all general (monotone and Lipschitz continuous) set reward functions $f(\mathbf{r}, S)$.

**Algorithm.** The algorithm is inspired by the recent work on misspecified contextual bandits (Lattimore et al., 2020). At time $t$, the algorithm constructs optimistic estimates $\widehat{r}_t(i)$ for all arms:

$$\widehat{r}_t(i) = \widehat{\theta}_t^\top \mathbf{x}_t(i) + \beta_t \sqrt{\mathbf{x}_t(i)\mathbf{V}_{t-1}^{-1}\mathbf{x}_t(i)}$$
$$+ \epsilon \sum_{s=1}^{t-1} \sum_{j \in S_s} \left| \mathbf{x}_t(i)\mathbf{V}_{t-1}^{-1}\mathbf{x}_s(j) \right| + \epsilon, \quad (2)$$

with $\widehat{\theta}_t = \mathbf{V}_{t-1}^{-1}\mathbf{b}_{t-1}$, $\mathbf{b}_{t-1} = \sum_{\tau=1}^{t-1} \sum_{i \in S_\tau} \widetilde{r}_\tau(i)\mathbf{x}_\tau(i)$, $\mathbf{V}_{t-1} = \lambda\mathbf{I} + \sum_{\tau=1}^{t-1} \sum_{i \in S_\tau} \mathbf{x}_\tau(i)\mathbf{x}_\tau(i)^\top$. Recall that $\widetilde{r}_\tau(i)$ is the observed stochastic score for arm $i$ at time step $\tau$. The online learner then queries the oracle $\mathcal{O}$ with the optimistic estimate $\widehat{\mathbf{r}}$ to obtain the set $S_t$. See Algorithm 1.

---

**Algorithm 1** MISSPECIFIED CONTEXTUAL COMBINATORIAL (MC$^2$) UCB FOR KNOWN $\epsilon$

1: **Input:** Misspecification level $\epsilon$, online ridge regression coefficient $\lambda$
2: **Initialize:** $\mathbf{V}_0 \leftarrow \lambda\mathbf{I}, \mathbf{b}_0 \leftarrow \mathbf{0}$
3: **for** $t = 1, \cdots, T$ **do**
4:     $\widehat{\theta}_t \leftarrow \mathbf{V}_{t-1}^{-1}\mathbf{b}_{t-1}$
5:     Calculate $\widehat{r}_t(i)$ according to Equation (2)
6:     Invoke oracle $\mathcal{O}$ with $\widehat{r}_t(i), i \in [m]$ to calculate set $S_t$ and play set $S_t$
7:     Observe the stochastic arms score $\{\widetilde{r}_t(i)\}_{i \in S_t}$ and reward $\widetilde{R}_t$
8:     Update $\mathbf{V}_t \leftarrow \mathbf{V}_{t-1} + \sum_{i \in S_t} \mathbf{x}_t(i)\mathbf{x}_t(i)^\top$ and $\mathbf{b}_t \leftarrow \mathbf{b}_{t-1} + \sum_{i \in S_t} \widetilde{r}_t(i)\mathbf{x}_t(i)$
9: **end for**

---

**Regret Analysis.** We now state our main result for the known $\epsilon$ case. Without loss of generality, assuming bounded parameters $\|\theta_*\|_2 \leq 1, \|\mathbf{x}_t(i)\|_2 \leq 1$ and bounded feedback $\widetilde{r}_t(i) \in [0, 1]$ for all $t \geq 0$ and $i \in [m]$, for any $0 < \delta < 1$, as one can rescale $\widetilde{r}_t(i)$ and $\mathbf{x}_t(i)$ to satisfy such assumptions. Setting $\beta_t = \sqrt{d \log(\frac{1+kt/\lambda}{\delta})} + \lambda^{1/2}$ and $\lambda > k$, we have the following regret bound.

**Theorem 5.1** (Regret Upper Bound for Algorithm 1). *With probability at least $1 - \delta$, the regret of Algorithm 1 is $R(T) \leq \widetilde{O}\left((\sqrt{d} + \sqrt{\lambda})\sqrt{kTd} + k\epsilon\sqrt{d}T\right)$, where $d$ is the dimension of features $\mathbf{x}_t(i)$, $k$ is the size of $S_t$.*

The $\widetilde{O}$ notation ignores the logarithmic terms. See Appendix B for the complete analysis. The regret bound has two parts: *well-specified bound* $\widetilde{O}\left((\sqrt{d} + \sqrt{\lambda})\sqrt{kTd}\right)$, which matches the established regret bound for well-specified models (Qin et al., 2014); and *misspecified bound* $\widetilde{O}\left(k\epsilon\sqrt{d}T\right)$, which nearly matches the $\Omega(k\epsilon\sqrt{d}T)$ lower bound in Proposition 4.1.

**Theorem 5.1 Proof Intuition:** The proof relies on showing that $\widehat{r}_t(i)$ is always an optimistic estimate of $r_t(i)$ despite the $\epsilon$ misspecification (Lemma B.2). Conceptually, this avoids under-estimating the arms belonging to the optimal set and converging to a sub-optimal set. Further, the regret of Algorithm 1 is closely related to the "tightness" of estimate $\widehat{r}_t(i)$, measured by $\sum_{t \in [T]} \sum_{t \in S_t} (\widehat{r}_t(i) - r_t^*(i))$. The summation of the first two terms in Equation (2) minus $r_t^*(i)$ can be bounded exactly as the $C^2UCB$ analysis, and the summation of the last two terms in Equation (2) gives the $\widetilde{O}(k\epsilon\sqrt{d}T)$ term.

# 6. Unknown Misspecification Level $\epsilon$

In this section, we present an algorithm and the corresponding regret bound that works without knowledge of the misspecification level $\epsilon$. Our proposed algorithm can be viewed

as a meta algorithm built upon base algorithms designed for the known $\epsilon$ case. Our theoretical analysis shows that an algorithm designed for known $\epsilon$ case can be coupled with the proposed meta algorithm to work in the unknown $\epsilon$ case, at the cost of a $\log^2 T$ multiplicative factor in the regret bound. We further instantiate the result with two different base-algorithm implementations to show the generality and the nearly optimal regret bound.

### 6.1. Algorithm

The algorithm for unknown misspecification level $\epsilon$ employs multiple base algorithms designed for the known $\epsilon$ case, with each of the base algorithms $\mathcal{A}_l$ having hypothetical misspecification level $\epsilon_l = \frac{2^l}{\sqrt{T}}$. The putative or promised regret bound of base algorithm $\mathcal{A}_l$ is typically in the form of:

$$Reg_{\mathcal{A}_l}(T) \leq \underbrace{Reg_0(T)}_{\textbf{Well-specified regret}} + \underbrace{\kappa\epsilon_l T}_{\textbf{Misspecified regret}}, \quad (3)$$

where the "**Well-specified regret**" $Reg_0(T)$ corresponds to the regret when there is no misspecification (e.g., $Reg_0(T) = \widetilde{O}\left((\sqrt{d} + \sqrt{\lambda})\sqrt{kTd}\right)$ for Algorithm 1); and the "**Misspecified regret**" $\kappa\epsilon_l T$ is induced by model misspecification (e.g., $\kappa = \widetilde{O}(k\sqrt{d})$ for Algorithm 1).

On top of the base algorithms, there is a meta algorithm that chooses one of the base algorithm $\mathcal{A}_l$ to construct and play set $S_t$ at each time step $t$. The meta algorithm conceptually treats each of the base algorithms $\mathcal{A}_l$ as an "arm", and adopts a UCB-like algorithm to choose $\mathcal{A}_l$. Though we are not the first to combine base bandits algorithms (see recent progress in (Pacchiano et al., 2020; Cutkosky et al., 2020; Arora et al., 2021)), our algorithm is specially designed for the misspecification problem and is therefore much simpler. Specifically, the optimistic estimate for base algorithm $\mathcal{A}_l$ is:

$$U(l,t) := \underbrace{\widehat{\mu}^l_{T(l,t-1)}}_{\textbf{Reward Avg.}} - \underbrace{\kappa\epsilon_l}_{\textbf{UCB shift}}$$
$$+ \underbrace{\min\left(1, \sqrt{\frac{32\log(T^3L/\delta)}{T(l,t-1)}}\right)}_{\textbf{Uncertainty of base algorithm } \mathcal{A}_l}. \quad (4)$$

The "**Reward Avg.**" term is the empirical reward average of base algorithm $\mathcal{A}_l$, defined as $\widehat{\mu}^l_{T(l,t-1)} := \frac{1}{T(l,t-1)}\sum_{\tau=1}^{t-1} \widetilde{R}_\tau \cdot \mathbb{I}(l_\tau = l)$, where $T(l,t-1) := \sum_{\tau=1}^{t-1} \mathbb{I}(l_\tau = l)$ is the number of times that $\mathcal{A}_l$ is invoked up to time step $t-1$. The "**Uncertainty of base algorithm** $\mathcal{A}_l$" corresponds to the uncertainty of the empirical reward average, where $L$ is the total number of base algorithms $\mathcal{A}_l$ and $\delta$ is some hyper-parameter corresponding to the failure

probability. The "**UCB shift**" term is inspired by (Cutkosky et al., 2020) and has two-fold implications: intuitively, the shift makes the base algorithm selection biased towards $\mathcal{A}_l$ with smaller hypothetical misspecification level $\epsilon_l$; technically, the shift makes the final regret bound depends only on $\epsilon_{l^*} := \min_l \epsilon_l \geq \epsilon$, instead of depending on all $\epsilon_l$. Note that $\epsilon$ is the true misspecification level unknown to the online learner.

Besides the typical UCB-like base algorithm selection, the meta algorithm is also equipped with base algorithm elimination - to eliminate the base algorithms with hypothetical misspecification level smaller than $\epsilon$. Any base algorithm $\mathcal{A}_l$ satisfying the following inequality will be eliminated:

$$\sum_{\tau=1}^{T(l,t)}\left(\widehat{\mu}^l_{\tau-1} - \widetilde{R}^l_\tau\right) \geq Reg_{\mathcal{A}_l}(T(l,t))$$
$$+ 3\sqrt{\log(T^3L/\delta)T(l,t)}. \quad (5)$$

Recall that $\widehat{\mu}^l_{\tau-1}$ is the empirical reward average of base algorithm $\mathcal{A}_l$ when it is invoked for $\tau-1$ times. $\widetilde{R}^l_\tau$ is the observed set reward when $\mathcal{A}_l$ is invoked for the $\tau$-th time. $Reg_{\mathcal{A}_l}(T(l,t))$ corresponds to Equation (3), which is the promised regret of base algorithm $\mathcal{A}_l$ played for $T(l,t)$ steps, if $\mathcal{A}_l$ has the correct misspecification level $\epsilon_l$ (i.e., $\epsilon_l \geq \epsilon$).

Intuitively, $\widehat{\mu}^l_{\tau-1}$ serves as an approximate proxy of $R^{l,*}_\tau$ (i.e., the optimal expected reward for the time step when $\mathcal{A}_l$ is invoked for its $\tau$-th time). The elimination criterion is then approximately comparing $\sum_{\tau=1}^{T(l,t)}\left(R^{l,*}_\tau - \widetilde{R}^l_\tau\right)$ with the promised regret bound $Reg_{\mathcal{A}_l}(T(l,t))$. If $\sum_{\tau=1}^{T(l,t)}\left(\widehat{\mu}^l_{\tau-1} - \widetilde{R}^l_\tau\right)$ is significantly larger than $Reg_{\mathcal{A}_l}(T(l,t))$, it implies that the base algorithm $\mathcal{A}_l$ has wrong hypothetical misspecification level $\epsilon_l$ and should be eliminated. See Algorithm 2 for the pseudocode.

### 6.2. Regret Analysis

We now present our main result for the unknown $\epsilon$ case. We again assume bounded parameters $\|\boldsymbol{\theta}_*\|_2 \leq 1$, $\|\mathbf{x}_t(i)\|_2 \leq 1$ and bounded feedback $\widetilde{r}_t(i) \in [0,1]$ for all $t \geq 0$ and $i \in [m]$.

**Theorem 6.1** (Regret Upper Bound for Algorithm 2). *With probability at least $1 - 4\delta$, the regret of Algorithm 2 is $R(T) \leq \left(1 + \frac{\log T}{2} + \log^2 T\right)Reg_{\mathcal{A}_{l^*}}(T) + (1 + 4\log T)\sqrt{8T\log(T^3L/\delta)}$, where $Reg_{\mathcal{A}_{l^*}}(T)$ is the regret bound for base algorithm $\mathcal{A}_{l^*}$, with $l^* = \arg\min_l \epsilon_l \geq \epsilon$.*

**Corollary 6.2.** *If the base algorithms $\{\mathcal{A}_l\}_l$ implement Algorithm 1, the regret of Algorithm 2 for the un-*

**Algorithm 2** MISSPECIFIED CONTEXTUAL COMBINATORIAL (MC$^2$) UCB FOR UNKNOWN $\epsilon$

1: **Input:** Time horizon $T$, online ridge regression coefficient $\lambda$
2: **Base Algorithms Initialization:** Base algorithms $\mathcal{A}_1, \cdots, \mathcal{A}_{\lceil \frac{\log T}{2} \rceil}$, where $\mathcal{A}_l$ implements Algorithm 1 with hypothetical misspecification level $\epsilon_l = \frac{2^l}{\sqrt{T}}$
3: **Initialize:** Set step counter $T(l,0) = 0$ and empirical reward average $\widehat{\mu}_0^l = 0$ for all base algorithms $\mathcal{A}_l$. Initialize active base algorithms set $\mathcal{L}_1 = \left\{ 1, \cdots, \lceil \frac{\log T}{2} \rceil \right\}$.
4: **for** $t = 1$ **to** $T$ **do**
5:     Calculate optimistic estimate $U(l,t)$ for all base algorithms $\mathcal{A}_l$ according to Equation (4)
6:     Choose base algorithm $\mathcal{A}_{l_t}$ with $l_t = \text{argmax}_{l \in \mathcal{L}_t} U(l,t)$
7:     Let base algorithm $\mathcal{A}_{l_t}$ choose and play a set $S_t$
8:     Observe the stochastic arms score $\{\widetilde{r}_t(i, S_t)\}_{i \in S_t}$ and reward $\widetilde{R}_t$.
9:     Send $\{\widetilde{r}_t(i, S_t)\}_{i \in S_t}$ to $\mathcal{A}_{l_t}$ for its updates
10:     Update $T(l_t, t) = T(l_t, t-1) + 1$ and $T(l, t) = T(l, t-1)$ for $l \neq l_t$
11:     Update $\widehat{\mu}_{T(l_t,t)}^{l_t} = \frac{1}{T(l_t,t)} \sum_{\tau=1}^{t} \widetilde{R}_\tau \cdot \mathbb{I}(l_\tau = l_t)$ and $\widehat{\mu}_{T(l,t)}^l = \widehat{\mu}_{T(l,t-1)}^l$ for $l \neq l_t$
12:     **if** elimination criterion Equation (5) is satisfied for $l_t$ **then**
13:         $\mathcal{L}_{t+1} = \mathcal{L}_t - \{l_t\}$
14:     **else**
15:         $\mathcal{L}_{t+1} = \mathcal{L}_t$
16:     **end if**
17: **end for**

known misspecification $\epsilon$ can be bounded as: $R(T) \leq \widetilde{O}\left( (\sqrt{d} + \sqrt{\lambda})\sqrt{kTd} + k\epsilon\sqrt{dT} \right)$.

Compared to the lower bound in Proposition 4.1, the regret bound in Corollary 6.2 is nearly optimal. Algorithm 2 is a general-purpose algorithm for leveraging multiple instances of a known misspecification base algorithm to obtain an algorithm for the unknown $\epsilon$ setting. In this context, we can extend our results to a setting where $r_t(i)$ is a general function of the context and the arm features (not just a linear function $\boldsymbol{\theta}_*^\top \mathbf{x}_t(i)$). The base algorithm can be Algorithm 1 in (Sen et al., 2021).

**Corollary 6.3.** *Consider the setting where $f(\mathbf{r}, S) = \sum_{i \in S} r_i$ and $r_t(i) = g^*(c_t, \mathbf{x}_t(i), i) + \Delta_t^{(i)}$ s.t. $\left| \Delta_t^{(i)} \right| < \epsilon$ (unknown to the learner). Suppose $g^*$ lies in a known finite function class $\mathcal{G}$. Then Algorithm 2 applied with Algorithm 1 in (Sen et al., 2021) as a base algorithm has a regret guarantee of, $R(T) \leq \widetilde{O}\left( k\sqrt{mT \log(|\mathcal{G}|)} + k\epsilon\sqrt{m}T \right)$, where $\widetilde{O}(\cdot)$ hides some polylog factors.*

The above corollary generalizes the result in (Sen et al., 2021) to unknown misspecification level $\epsilon$. We provide a more detailed result as Corollary C.1 in the appendix. Note that as noted in (Sen et al., 2021), the above result can be generalized to efficiently learnable infinite function classes.

**Theorem 6.1 Proof Intuition:** The interesting part is to prove the overall regret only depends on $\epsilon_{l^*}$, where $l^* := \text{argmin}_l \epsilon_l \geq \epsilon$. Consider step $t$ and let $\mathcal{A}_{l_t}$ be the invoked base algorithm. The single step regret can be rewritten as:

$$R_t^* - R_{T(l_t,t)}^{l_t} = \underbrace{R_t^* - U(l^*,t)}_{(a)} + \underbrace{U(l^*,t) - U(l_t,t)}_{(b)}$$
$$+ \underbrace{U(l_t,t) - \widehat{\mu}_{T(l_t,t-1)}^{l_t}}_{(c)}$$
$$+ \underbrace{\widehat{\mu}_{T(l_t,t-1)}^{l_t} - R_{T(l_t,t)}^{l_t}}_{(d)}$$

Intuitively, **(a)** only depends on $l_*$. As $l_t$ is selected at time $t$, **(b)** is smaller than 0 and therefore is ignored in the upper bound. **(c)** has a $-\kappa\epsilon_l$ term, which originates from the **UCB shift** term in the $U(l,t)$ definition (Equation (4)). Finally, **(d)** has a $\kappa\epsilon_{l_t}$ term, since $\mathcal{A}_{l_t}$ has a $\kappa\epsilon_{l_t}T$ misspecified regret. Note that the $\kappa\epsilon_{l_t}$ in **(c)** and **(d)** cancels. The regret $R(T)$ therefore only depends on $\epsilon_{l^*}$.

## 7. Experiments

In this section, we evaluate the proposed algorithms on a synthetic environment and an environment derived from a real-world dataset. The results show that our proposed algorithm (MC$^2$UCB) is significantly better than C$^2$UCB (Qin et al., 2014) (which is near-optimal for the well-specified contextual combinatorial bandits) when the underlying environment is misspecified. Further, MC$^2$UCB closely matches the performance of C$^2$UCB when the underlying environment is nearly well-specified.

**Synthetic Experiments.** The synthetic environment is constructed in the following way: We first generate 100 arms with features $\mathbf{x}(i)$ and the vector $\boldsymbol{\theta}_*$ from a 10-dimensional spherical Gaussian distribution $\mathcal{N}(0, \frac{1}{\sqrt{10}}I_{10})$. For the well-specified linear model, the score for the arm $i$ at time step $t$ is generated as $r(i) = \mathbf{x}(i)^\top \boldsymbol{\theta}_* + \eta_t$, where $\eta_t \sim \mathcal{N}(0, 0.1)$ is noise. The goal of the online learner is to find the optimal set of 3 arms, which maximizes the sum of expected scores.

To create misspecification, we first rank the arms according to the inner product $\mathbf{x}(i)^\top \boldsymbol{\theta}_*$ in descending order and let $\mathcal{B}$ denote the set of arms with indices in $\{30, \cdots, 39\}$. Intuitively, $\mathcal{B}$ is a set of bad arms in the well-specified environment. We change their scores to be $r(i) = \mathbf{x}(i)^\top \boldsymbol{\theta}_* + \epsilon + \eta_t, \forall i \in \mathcal{B}$, where $\epsilon$ controls the level of misspecification. Specifically, we set $\epsilon = 2$, which completely changes the the optimal set when comparing to the one under a well-specified linear model $\boldsymbol{\theta}_*$.

We evaluate three algorithms: "MC$^2$UCB with $\epsilon = 2$" correspond to Algorithm 1 with $\epsilon = 2$. "MC$^2$UCB with unknown $\epsilon$" corresponds to Algorithm 2, which adopts 3 base

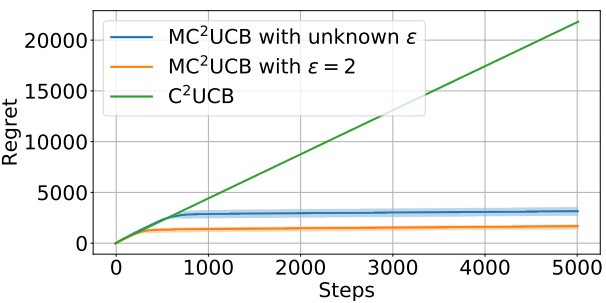

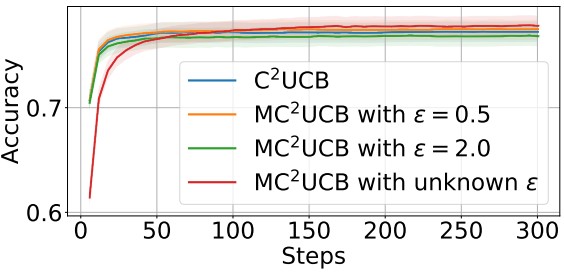

(a) No artificial misspecification

Figure 1: Performance of different learning algorithms in a misspecified environment. $MC^2UCB$ with known $\epsilon$ (Algorithm 1) and unknown $\epsilon$ (Algorithm 2) significantly outperforms the $C^2UCB$. More importantly, $C^2UCB$ converges to the wrong set and has a linear regret. This shows that the $MC^2UCB$ algorithms can well adapt to misspecification and avoid catastrophic failure.

algorithms with hypothetical $\epsilon$ in $\{0, 0.5, 2\}$. "$C^2UCB$" implements the Contextual Combinatorial UCB (Qin et al., 2014), which is designed for the well-specified contextual bandits. The average regret (and standard deviation) of 5 runs are reported in Figure 1.

**Application to Online Movie Recommendation.** We also evaluate the performance of our proposed algorithms on a real-world problem. The environment is derived from a popular recommendation dataset: Movielens-1M, which contains over 1 million ratings of 3952 movies by 6040 users.

We construct the environment similar to (Qin et al., 2014). Specifically, we split the users into a training set (containing 5740 users) and a test set (containing 300 users with more than 100 ratings). We then use a rank-16 matrix factorization on the ratings from the training set to create movies' features. Those movie features are used as arms' features ($\mathbf{x}_t(i)$) in the online learning environment.

For each of the users in the testing set, the online learning algorithm recommends 30 movies to the user in every time step $t$. A movie with a rating of 4 or 5 will have a score of 1, and other movies have a score of 0. The algorithm interacts with the user for 150 times (i.e., $T = 150$). The performance is measured by recommendation accuracy: $\frac{|\mathcal{P} \cap S_t|}{|S_t|}$, where $\mathcal{P}$ is the set of movies that the user gives rating 4 or 5.

For the misspecified environments, we consider the case where the first 25 feedback come a different model: The environment draws a $\theta'$ from spherical Gaussian, and the movies whose embedding has top-20 inner product with $\theta'$ have score 1 and all others have score 0. $\theta'$ is fixed for the first 25 steps. The rewards of the entire time horizon thus do not follow a single linear model.

We test $C^2UCB$ and $MC^2UCB$ on both environments (i.e.,

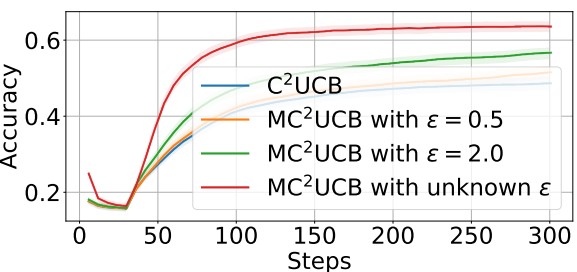

(b) First 25 steps misspecified

Figure 2: In the "No artificial misspecification" environment (Fig. (a)), the $MC^2UCB$ with known $\epsilon = 0.5$ and $MC^2UCB$ with unknown misspecification yield slightly better performances, which shows that the $MC^2UCB$ algorithms can well adapt to the misspecification that naturally arises in a real-world dataset. Further, in the misspecified environment (Fig. (b)), $MC^2UCB$ algorithms offer significantly higher accuracy.

with/without artificial misspecification). We set $\epsilon = 0.5$ or 2 for $MC^2UCB$ with known misspecification, and set $MC^2UCB$ with unknown to adopt 3 base algorithms with hypothetical $\epsilon$ in $\{0, 0.5, 2\}$. The results in Figure 2 are the average and standard deivation of 300 testing users. The result shows the superior performance of $MC^2UCB$, as it significantly outperforms the $C^2UCB$ when the model is misspecified and loses nothing in the nearly well-specified environment.

## Acknowledgement

This work is supported by NSF grant 1934932.

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

# A. Proof of Section 4

## A.1. Proof of Proposition 4.1

*Proof.* We begin by construct $m$ unit $l_2$ norm points in $\mathbb{R}^d$, denote the stacking of those points as $\mathbf{X}_0 \in \mathbb{R}^{m \times d}$. We construct $\mathbf{X}_0$ such that

$$\mathbf{a}^\top \mathbf{a} = 1, \ \forall \mathbf{a} \in \text{rows}(\mathbf{X}_0) \quad \text{and} \quad \mathbf{a}^\top \mathbf{b} \le \sqrt{\frac{8 \log(m)}{d-1}}, \ \forall \mathbf{a}, \mathbf{b} \in \text{rows}(\mathbf{X}_0).$$

The existence of such construction follows from Lemma 3.1 of (Lattimore et al., 2020). Replicate $\mathbf{X}_0$ for $k$ times, denote the resulting matrix to be $\mathbf{X} \in \mathbb{R}^{km \times d}$. We next prove the lower bound with the bandits problem instance with arms' features being $\mathbf{X}$.

Choose $\mathbf{a}^* \in \text{rows}(\mathbf{X})$ and let

$$\boldsymbol{\theta}_* = \epsilon \sqrt{\frac{d-1}{8 \log(m)}} \mathbf{a}^*.$$

Further, let the expected score of arms be

$$r_t(\mathbf{a}^*) = \boldsymbol{\theta}_*^\top \mathbf{a}^* = \epsilon \quad \text{and} \quad r_t(\mathbf{b}) = \boldsymbol{\theta}_*^\top \mathbf{b} + \Delta_t(\mathbf{b}) = 0 \ \forall \mathbf{b} \ne \mathbf{a}^*.$$

It is easy to verify that $|\Delta_t(\mathbf{a})| \le \epsilon, \forall \mathbf{a} \in \text{rows}(\mathbf{X}), t \in [T]$. Let $\tau = \min \{t \le T : \mathbf{a}^* \in S_t\}$, i.e., the first time $S_t$ contains $\mathbf{a}^*$. Notice that the optimal set $S^*$, composed by $k$ copies of $\mathbf{a}^*$, has reward $k\epsilon \sqrt{\frac{d-1}{8 \log(m)}}$. Also notice that all arms other than $\mathbf{a}^*$ gives 0 reward. We have that

$$R(T) = k\epsilon \sqrt{\frac{d-1}{8 \log(m)}} \mathbb{E}[\tau].$$

Notice that any there are $m$ different features in $\mathbf{X}$, and the algorithm sees the reward for at most $k$ arms in each time step. Since the arm $\mathbf{a}^*$ is randomly chosen from $\mathbf{X}$ and the feedback of all other arms are 0 (which therefore provides no information about $\mathbf{a}^*$), we have that $\mathbb{E}[\tau] \ge \min(T, \frac{m}{2k})$. Therefore the regret lower bound is

$$R(T) = k\epsilon \sqrt{\frac{d-1}{8 \log(m)}} \min\left(T, \frac{m}{2k}\right).$$

This completes the proof. ∎

## A.2. Proof of Proposition 4.3

The following proof is inspired by "Failure of unmodified algorithm" in Appendix E of (Lattimore et al., 2020).

*Proof.* Consider the simple case with feature dimension $d = 2$. Let the arms' features $\{\mathbf{x}_t(1), \cdots, \mathbf{x}_t(m)\}$ be that

- For $t \le T/2$ and $t$ is odd, $\mathbf{x}_t(i) = [\epsilon, 0], \forall i \in [m]$.
- For $t \le T/2$ and $t$ is even, $\mathbf{x}_t(i) = [0, \epsilon], \forall i \in [m]$.
- For $t > T/2$, $\mathbf{x}_t(i) = [1, 0], \forall i \le m/2$ and $\mathbf{x}_t(i) = [\epsilon, \epsilon] \forall i > m/2$.

Further, we assume $m > 2k$ and no noise. Let $\theta^* = [\frac{1}{2}, -\frac{1}{2}]$. For the misspecification, let

$$\Delta_t([\epsilon, 0]) = -\epsilon, \quad \Delta_t([0, \epsilon]) = \epsilon, \quad \Delta_t([1, 0]) = 0, \quad \Delta_t([\epsilon, \epsilon]) = 0.$$

Notice that for $t = T/2$, the algorithm has 0 regret and

$$\mathbf{V}_{T/2} = \begin{bmatrix} \lambda + \frac{kT\epsilon^2}{4} & 0 \\ 0 & \lambda + \frac{kT\epsilon^2}{4} \end{bmatrix}, \quad \mathbf{b}_{T/2} = \begin{bmatrix} -\frac{kT\epsilon^2}{4} \\ \frac{kT\epsilon^2}{4} \end{bmatrix}$$

therefore $\widehat{\boldsymbol{\theta}}_{T/2} = \left[ -\frac{kT\epsilon^2}{4\lambda + kT\epsilon^2}, \frac{kT\epsilon^2}{4\lambda + kT\epsilon^2} \right]$. For $t = T/2 + 1$, we have

$$\widehat{\boldsymbol{\theta}}_t^\top [1, 0] + \beta_t \|[1, 0]\|_{\mathbf{V}_t^{-1}} = -\frac{kT\epsilon^2}{4\lambda + kT\epsilon^2} + O\left( \sqrt{\frac{d \log T}{kT\epsilon^2}} \right)$$

and

$$\widehat{\boldsymbol{\theta}}_t^\top [\epsilon, \epsilon] + \beta_t \|[\epsilon, \epsilon]\|_{\mathbf{V}_t^{-1}} = 0 + O\left( \sqrt{\frac{d \log T}{kT}} \right)$$

Therefore for $kT\epsilon^2$ large enough (i.e., larger than some problem independent constant $C_0$), the algorithm will choose $k$ arms with feature $[\epsilon, \epsilon]$ to construct the set.

We next show that the algorithm will play the set composed by $k$ arms with feature $[\epsilon, \epsilon]$ until $T$. Suppose the algorithm has played such set up to time $t_0 > T/2$, then

$$\mathbf{V}_{t_0} = \begin{bmatrix} \lambda + \frac{kT\epsilon^2}{4} + k(t_0 - T/2)\epsilon^2 & k(t_0 - T/2)\epsilon^2 \\ k(t_0 - T/2)\epsilon^2 & \lambda + \frac{kT\epsilon^2}{4} + k(t_0 - T/2)\epsilon^2 \end{bmatrix}, \quad \mathbf{b}_{t_0} = \begin{bmatrix} -\frac{kT\epsilon^2}{4} \\ \frac{kT\epsilon^2}{4} \end{bmatrix}$$

and again we have $\widehat{\boldsymbol{\theta}}_{t_0} = \left[ -\frac{kT\epsilon^2}{4\lambda + kT\epsilon^2}, \frac{kT\epsilon^2}{4\lambda + kT\epsilon^2} \right]$ and for $t = t_0 + 1$

$$\widehat{\boldsymbol{\theta}}_t^\top [1, 0] + \beta_t \|[1, 0]\|_{\mathbf{V}_t^{-1}} = -\frac{kT\epsilon^2}{4\lambda + kT\epsilon^2} + O\left( \sqrt{\frac{d \log T}{4kt_0\epsilon^2 - kT\epsilon^2}} \right)$$

and

$$\widehat{\boldsymbol{\theta}}_t^\top [\epsilon, \epsilon] + \beta_t \|[\epsilon, \epsilon]\|_{\mathbf{V}_t^{-1}} > 0$$

Therefore the algorithm will continue play the set composed by $k$ copies of $[\epsilon, \epsilon]$.

Notice that given the problem instance construction, we have $\mathbf{x}_t(i) = [1, 0]$ has $1/2$ reward and $\mathbf{x}_t(i) = [\epsilon, \epsilon]$ has $0$ reward. Therefore for $t > T/2$, the algorithm induces $kT/2$ regret. This compeltes the proof.

$\blacksquare$

## B. Proof of Section 5

We first present the complete version of Theorem B.1 which provides an explicit regret bound.

**Theorem B.1** (Regret upper bound of Algorithm 2). *Assume that $\|\theta_*\|_2 \leq 1$, $\|\mathbf{x}_t(i)\|_2 \leq 1$ and $r_t(i) \in [0, 1]$ for all $t \geq 0$ and $i \in [m]$. Given $0 < \delta < 1$, set $\beta_t = \sqrt{d \log(\frac{1+kt/\lambda}{\delta})} + \lambda^{1/2}$ with $\lambda \geq k$. Then, with probability at least $1 - \delta$, the total regret can be bounded as*

$$R(T) \leq 2C \left( \sqrt{d \log \left( \frac{1 + kt/\lambda}{\delta} \right)} + \lambda^{1/2} \right) \sqrt{kTd \log \left( 1 + \frac{Tk}{\lambda d} \right)}$$

$$+ 2\epsilon k C T \sqrt{d \log \left( 1 + \frac{Tk}{\lambda d} \right)} + 2\epsilon k C T.$$

*Ignoring the constants and logarithm terms, the regret bound can be simplified as*

$$R(T) = \widetilde{O} \left( (\sqrt{d} + \sqrt{\lambda})\sqrt{kTd} + k\epsilon\sqrt{d}T \right).$$

**Lemma B.2.** *If we set $\beta_t = \sqrt{d \log \left( \frac{1+kt/\lambda}{\delta} \right)} + \lambda^{1/2}$, with probability at least $1 - \delta$, for all $S$ and $i \in S$, we have*

$$0 \leq \widehat{r}_t(i) - r_t^*(i, S) \leq 2\beta_t \|\mathbf{x}_t(i)\|_{\mathbf{V}_{t-1}^{-1}} + 2\epsilon \sum_{s=1}^{t-1} \sum_{j \in S_s} \left| \mathbf{x}_t(i) \mathbf{V}_{t-1}^{-1} \mathbf{x}_s(j) \right| + 2\epsilon.$$

*Proof.* Notice that

$$
\begin{aligned}
\widehat{\boldsymbol{\theta}}_t =& \mathbf{V}_{t-1}^{-1}\left(\sum_{s=1}^{t-1}\sum_{i\in S_s}\mathbf{x}_s(i)\cdot\widetilde{r}_s(i,S_s)\right)\\
=& \mathbf{V}_{t-1}^{-1}\left(\sum_{s=1}^{t-1}\sum_{i\in S_s}\mathbf{x}_s(i)\cdot\left(\boldsymbol{\theta}_*^\top\mathbf{x}_s(i)+\eta_s(i)+\Delta_s(i,S_s)\right)\right)\\
=& \widehat{\boldsymbol{\theta}}_t^+ + \mathbf{V}_{t-1}^{-1}\sum_{s=1}^{t-1}\sum_{i\in S_s}\mathbf{x}_s(i)\cdot\Delta_s(i,S_s),
\end{aligned}
$$

where we define $\widehat{\boldsymbol{\theta}}_t^+ := \mathbf{V}_{t-1}^{-1}\left(\sum_{s=1}^{t-1}\sum_{i\in S_s}\mathbf{x}_s(i)\cdot\left(\boldsymbol{\theta}_*^\top\mathbf{x}_s(i)+\eta_s(i)\right)\right)$. Therefore, we have

$$
\widehat{r}_t(i)-r_t^*(i,S) = \underbrace{\left(\widehat{\boldsymbol{\theta}}_t^+ - \boldsymbol{\theta}_*\right)^\top\mathbf{x}_t(i)+\beta_t\sqrt{\mathbf{x}_t(i)\mathbf{V}_{t-1}^{-1}\mathbf{x}_t(i)}}_{\text{term (a)}}
$$

$$
+ \underbrace{\mathbf{x}_t(i)^\top\mathbf{V}_{t-1}^{-1}\sum_{s=1}^{t-1}\sum_{j\in S_s}\mathbf{x}_s(j)\cdot\Delta_s(j,S_s)+\epsilon\sum_{s=1}^{t-1}\sum_{j\in S_s}\left|\mathbf{x}_t(i)\mathbf{V}_{t-1}^{-1}\mathbf{x}_s(j)\right|}_{\text{term (b)}}
$$

$$
+ \underbrace{\epsilon-\Delta_t(i,S)}_{\text{term (c)}}.
$$

To see the range of term (a), applying Theorem 2 of (Abbasi-Yadkori et al., 2011) (we provided it in Lemma B.3) and we have

$$
\begin{aligned}
\left(\widehat{\boldsymbol{\theta}}_t^+ - \boldsymbol{\theta}_*\right)^\top\mathbf{x}_t(i)+\beta_t\sqrt{\mathbf{x}_t(i)\mathbf{V}_{t-1}^{-1}\mathbf{x}_t(i)} &\leq \left(\|\widehat{\boldsymbol{\theta}}_t^+ - \boldsymbol{\theta}_*\|_{\mathbf{V}_{t-1}}+\beta_t\right)\|\mathbf{x}_t(i)\|_{\mathbf{V}_{t-1}^{-1}}\\
&\leq 2\beta_t\|\mathbf{x}_t(i)\|_{\mathbf{V}_{t-1}^{-1}}.
\end{aligned}
$$

Similarly we can show the term (a) is bounded below by 0.

For term (b), notice that

$$
\left|\mathbf{x}_t(i)^\top\mathbf{V}_{t-1}^{-1}\sum_{s=1}^{t-1}\sum_{j\in S_s}\mathbf{x}_s(j)\cdot\Delta_s(j,S_s)\right| \leq \epsilon\sum_{s=1}^{t-1}\sum_{j\in S_s}\left|\mathbf{x}_t(i)\mathbf{V}_{t-1}^{-1}\mathbf{x}_s(j)\right|,
$$

where the inequality follows as all $\Delta_s(j,S_s)$s are bounded by $\epsilon$. Therefore the second term is bounded below by 0 and upper bounded by $2\epsilon\sum_{s=1}^{t-1}\sum_{j\in S_s}\left|\mathbf{x}_t(i)\mathbf{V}_{t-1}^{-1}\mathbf{x}_s(j)\right|$.

Lastly, one can easily verify that the last term ranges from 0 to $2\epsilon$. ∎

**Lemma B.3.** *For $\|\theta_*\|_2 \leq 1$ and $\|\mathbf{x}_t(i)\|_2 \leq 1, \forall t \in [T]$ and $i \in [m]$, define $\mathbf{V}_t = \lambda\mathbf{I}+\sum_{s=1}^{t}\sum_{i\in S_s}\mathbf{x}_t(i)\mathbf{x}_t(i)^\top$. Define $\widehat{\boldsymbol{\theta}}_t^+ := \mathbf{V}_{t-1}^{-1}\left(\sum_{s=1}^{t-1}\sum_{i\in S_s}\mathbf{x}_s(i)\cdot\left(\boldsymbol{\theta}_*^\top\mathbf{x}_s(i)+\eta_s(i)\right)\right)$, we have*

$$
\|\widehat{\boldsymbol{\theta}}_t^+ - \boldsymbol{\theta}_*\|_{\mathbf{V}_{t-1}} \leq \sqrt{d\log\left(\frac{1+kt/\lambda}{\delta}\right)}+\lambda^{1/2}.
$$

*Proof.* This follows directly from Theorem 2 of (Abbasi-Yadkori et al., 2011). ∎

**Lemma B.4** (Lemma 4.2 of (Qin et al., 2014)). *Let $V_t = \lambda I+\sum_{s=1}^{t}\sum_{i\in S_s}\mathbf{x}_s(i)\mathbf{x}_s(i)^\top$. For $\lambda \geq k$, we have*

$$
\sum_{s=1}^{t}\sum_{i\in S_s}\|\mathbf{x}_s(i)\|_{\mathbf{V}_{t-1}^{-1}}^2 \leq 2d\log\left(1+\frac{tk}{\lambda d}\right).
$$

Now we present the proof of Theorem B.1.

*Proof.* We first bound the regret at round $t$ as follows,

$$Reg_t = f(r_t^*, S^*) - f(r_t^*, S_t) \leq f(\widehat{r}_t, S_t) - f(r_t^*, S_t) \leq C\sqrt{\sum_{i \in S_t} (\widehat{r}_t(i) - r_t^*(i))^2}$$

$$\leq C\sum_{i \in S_t} (\widehat{r}_t(i) - r_t^*(i)).$$

Therefore with probability at least $1 - \delta$,

$$R(T) \leq C\sum_{t=1}^{T}\sum_{i \in S_t} (\widehat{r}_t(i) - r_t^*(i))$$

$$\leq C\sum_{t=1}^{T}\sum_{i \in S_t} 2\beta_t \|\mathbf{x}_t(i)\|_{\mathbf{V}_{t-1}^{-1}} + 2\epsilon C\sum_{t=1}^{T}\sum_{i \in S_t}\sum_{s=1}^{t-1}\sum_{j \in S_s} \left|\mathbf{x}_t(i)\mathbf{V}_{t-1}^{-1}\mathbf{x}_s(j)\right| + 2\epsilon kCT$$

$$\leq C\sqrt{kT\sum_{t=1}^{T}\sum_{i \in S_t} 4\beta_t^2 \|\mathbf{x}_t(i)\|_{\mathbf{V}_{t-1}^{-1}}^2}$$

$$+ 2\epsilon kCT\sqrt{\sum_{t=1}^{T}\sum_{i \in S_t}\sum_{s=1}^{t-1}\sum_{j \in S_s} \left(\mathbf{x}_t(i)\mathbf{V}_{t-1}^{-1}\mathbf{x}_s(j)\right)^2} + 2\epsilon kCT$$

$$\leq 2C\beta_T\sqrt{kT\sum_{t=1}^{T}\sum_{i \in S_t} \|\mathbf{x}_t(i)\|_{\mathbf{V}_{t-1}^{-1}}^2} + 2\epsilon kCT\sqrt{\sum_{t=1}^{T}\sum_{i \in S_t} \|\mathbf{x}_t(i)\|_{\mathbf{V}_{t-1}^{-1}}^2} + 2\epsilon kCT.$$

With $\beta_T = \sqrt{d\log\left(\frac{1+kT/\lambda}{\delta}\right)} + \lambda^{1/2}$ and Lemma B.4, we have

$$R(T) \leq 2C\left(\sqrt{d\log\left(\frac{1+kT/\lambda}{\delta}\right)} + \lambda^{1/2}\right)\sqrt{kTd\log\left(1 + \frac{Tk}{\lambda d}\right)}$$

$$+ 2\epsilon kCT\sqrt{d\log\left(1 + \frac{Tk}{\lambda d}\right)} + 2\epsilon kCT.$$

Ignoring the log terms, we have $R(T) = \widetilde{O}\left(C(\sqrt{d} + \sqrt{\lambda})\sqrt{kTd} + k\epsilon\sqrt{dT}\right)$ ∎

## C. Deferred Results of Section 6

### C.1. Formal Version of Corollary 6.3

**Corollary C.1.** *Consider the setting where $f(\mathbf{r}, S) = \sum_{i \in S} r_i$ and $r_t(i) = g^*(c_t, \mathbf{x}_t(i), i) + \Delta_t^{(i)}$ s.t. $\left|\Delta_t^{(i)}\right| < \epsilon$ (unknown to the learner). Suppose $g^*$ lies in a known finite function class $\mathcal{G}$. Then Algorithm 2 applied with Algorithm 1 in (Sen et al., 2021) as a base algorithm has a regret guarantee of,*

$$R(T) \leq C\left(1 + \frac{\log T}{2} + \log^2 T\right)\left(k\sqrt{(m - k + 1)T\log\left(\frac{|\mathcal{G}|T}{\delta}\right)} + \epsilon kT\sqrt{m - k + 1}\right)$$

$$+ (1 + 4\log T)\sqrt{8T\log(T^3 L/\delta)}.$$

### C.2. Proof of Theorem 6.1

We first present the complete version of Theorem 6.1 which proivdes an explicit regret bound.

**Theorem C.2** (Regret upper bound of Algorithm 2). *Assume that $\|\theta_*\| \leq B$, $\|\mathbf{x}_t(i)\|_2 \leq 1$ and $r_t(i) \in [0,1]$ for all $t \geq 0$ and $i \in [m]$. Then with probability at least $1 - 4\delta$, the regret can be bounded as*

$$R(T) \leq \left( 1 + \frac{\log T}{2} + \log^2 T \right) Reg_{\mathcal{A}_{l^*}}(T) + (1 + 5 \log T) \sqrt{8T \log(T^3 L/\delta)},$$

*where $Reg_{\mathcal{A}_{l^*}}(T)$ is the regret bound from Algorithm 1 for base algorithm $\mathcal{A}_{l^*}$, with $l^* = \arg\min_l \epsilon_l \geq \epsilon$. Ignoring the constants and log terms, we have*

$$R(T) = \widetilde{O}\left( (\sqrt{d} + \sqrt{\lambda})\sqrt{kTd} + k\epsilon\sqrt{d}T \right).$$

**Notation in the proof.** We denote $\widetilde{R}_\tau^l$ as the observed stochastic reward when the meta algorithm chooses base algorithm $\mathcal{A}_l$ for the $\tau$-th time, and denote its expectation to be $R_\tau^l$. For that time step, we denote $R_\tau^{l,*}$ to be the expectation of the optimal reward. Further, we denote $T(l,t)$ to be the number of times we pick base algorithm $\mathcal{A}_l$ up to time step $t$.

Let $\epsilon$ be the true misspecification bound, and define $l^* = \arg\min_l \epsilon_l \geq \epsilon$. Before proving Theorem 6.1, we first present a few lemmas.

**Lemma C.3** (Adapted from Lemma 8 of (Cutkosky et al., 2020)). *With probability at least $1 - \delta$, for all $l$ and $t$, we have*

$$\sum_{\tau=1}^{T(l,t)} \left| R_\tau^l - \widetilde{R}_\tau^l \right| \leq \sqrt{8T(l,t) \log(T^3 L/\delta)}.$$

*Proof.* Notice that $R_\tau^l$ is the expectation of $\widetilde{R}_\tau^l$. This follows directly from a martingale-type concentration. ∎

**Lemma C.4** (Adapted from Lemma 9 of (Cutkosky et al., 2020)). *With probability at least $1 - \delta$, for all $l$, for all $t$, we have*

$$\sum_{\tau=1}^{T(l,t)} \widehat{\mu}_\tau^l - R_\tau^{l,*} \leq 3\sqrt{8T(l,t) \log(T^3 L/\delta)}.$$

*Proof.* By a martingale-type concentration, for all $l$ and $t$, we have

$$\sum_{\tau=1}^{T(l,t)} \left( \widetilde{R}_\tau^l - \mathbb{E}_{c_\tau}\left[ R_\tau^l \right] \right) \leq \sqrt{8T(l,t) \log(T^3 L/\delta)}.$$

Define $R_* = \mathbb{E}_{c_t}[R_t^*]$, where the expectation is taken over the context $c_t \sim \mathcal{D}_\mathcal{C}$. For any context $c_t$, we have $R_t^l \leq R_t^*$. Thus we have $\mathbb{E}_{c_t}[R_t^l] \leq \mathbb{E}_{c_t}[R_t^*]$, and

$$\sum_{\tau=1}^{T(l,t)} \left( \widetilde{R}_\tau^l - R_* \right) \leq \sqrt{8T(l,t) \log(T^3 L/\delta)}.$$

Therefore for any $l$ and $t$, we have

$$\widehat{\mu}_t^l - R_* \leq \sqrt{\frac{8 \log(T^3 L/\delta)}{T(l,t)}} \implies \sum_{\tau=1}^{T(l,t)} \widehat{\mu}_\tau^l - R_* \leq 2\sqrt{8T(l,t) \log(T^3 L/\delta)}. \tag{6}$$

Further notice that $R_\tau^{l,*}$ is a random variable in $[0,1]$, with expectation being $R_*$ and randomness induced by the context $c_t$, we have that

$$\sum_{\tau=1}^{T(l,t)} R_* - R_\tau^{l,*} \leq \sqrt{\frac{\log(\frac{1}{\delta})T(l,t)}{2}} \leq \sqrt{8 \log(T^3 L/\delta)T(l,t)}. \tag{7}$$

The first inequality follows from Hoeffding's inequality. Combining Equations (6) and (7) completes the proof. ∎

Recall that $R_* = \mathbb{E}_{c_t}[R_t^*]$, i.e., taking expectation over the context $c_t \sim \mathcal{D}_{\mathcal{C}}$. We have the following lemma.

**Lemma C.5** (Simplified Lemma 10 of (Cutkosky et al., 2020)). *With probability at least $1 - 4\delta$, for $l^* = \arg\min_l \epsilon_l \geq \epsilon$ and for all $t$, we have the following hold simultaneously*

$$\sum_{\tau=1}^{T(l^*,t)} R_\tau^{l^*,*} - R_\tau^{l^*} \leq Reg_{\mathcal{A}_{l^*}}(T(l^*,t)),$$

$$R_* - \widehat{\mu}_{T(l^*,t-1)}^{l^*} \leq \min\left(1, \frac{Reg_{\mathcal{A}_{l^*}}(T(l^*,t-1))}{T(l^*,t-1)} + 2\sqrt{\frac{8\log(T^3 L/\delta)}{T(l^*,t-1)}}\right),$$

$$\sum_{\tau=1}^{T(l^*,t-1)} \widehat{\mu}_{\tau-1}^{l^*} - \widetilde{R}_\tau^{l^*} \leq Reg_{\mathcal{A}_l^*}(T(l^*,t-1)) + 4\sqrt{8T(l^*,t-1)\log(T^3 L/\delta)}.$$

*Proof.* With Theorem B.1, with probability $1 - \delta$, we have that for all $t$, we have

$$\sum_{t=1}^{T(l^*,t)} R_\tau^{l^*,*} - R_\tau^{l^*} \leq Reg_{\mathcal{A}_{l^*}}(T(l^*,t)).$$

With Lemma C.3, with probability $1 - \delta$, for all $t$, we have

$$\sum_{\tau=1}^{T(l^*,t)} R_\tau^{l^*} - \widetilde{R}_\tau^{l^*} \leq \sqrt{8T(l^*,t)\log(T^3 L/\delta)}.$$

With Lemma C.4, with probability $1 - \delta$, for all $t$, we have

$$\sum_{\tau=1}^{T(l^*,t)} \widehat{\mu}_\tau^{l^*} - R_\tau^{l^*,*} \leq 3\sqrt{8T(l^*,t)\log(T^3 L/\delta)}.$$

With Hoeffding's inequality, for all $l \in [L]$ and all $t$, with probability at least $1 - \delta$, we have

$$\sum_{\tau=1}^{T(l,t-1)} R_* - R_\tau^{l,*} \leq \sqrt{8T(l,t)\log(T^3 L/\delta)}$$

The rest of proof conditions on the above 4 equations to hold, which has probability at least $1 - 4\delta$. Note that the first claim in the lemma immediately follows from the first equation above.

For the second equation, the 1 of the minimum holds by assumption that $R_* \leq 1$. For $T(l^*,t-1) \neq 0$, we have

$$R_* - \widehat{\mu}_{T(l^*,t-1)}^{l^*} = R_* - \frac{1}{T(l^*,t-1)}\sum_{\tau=1}^{T(l^*,t-1)} R_\tau^{l^*,*} + \frac{1}{T(l^*,t-1)}\sum_{\tau=1}^{T(l^*,t-1)}\left(R_\tau^{l^*,*} - \widetilde{R}_\tau^{l^*}\right)$$

$$\leq \frac{Reg_{\mathcal{A}_{l^*}}(T(l^*,t-1))}{T(l^*,t-1)} + 2\sqrt{\frac{8\log(T^3 L/\delta)}{T(l^*,t-1)}}.$$

For the third equation, we have

$$\sum_{\tau=1}^{T(l^*,t-1)} \widehat{\mu}_{\tau-1}^{l^*} - \widetilde{R}_\tau^{l^*} = \sum_{\tau=1}^{T(l^*,t-1)} \widehat{\mu}_{\tau-1}^{l^*} - R_\tau^{l^*,*} + R_\tau^{l^*,*} - R_\tau^{l^*} + R_\tau^{l^*} - \widetilde{R}_\tau^{l^*}$$

$$\leq Reg_{\mathcal{A}_{l^*}}(T(l^*,t-1)) + 4\sqrt{8T(l^*,t-1)\log(T^3 L/\delta)}.$$

∎

**Lemma C.6** (Adapted from Lemma 11 of (Cutkosky et al., 2020)). *With probability at least $1 - \delta$, all base algorithms $\mathcal{A}_l$ satisfy:*

$$\sum_{\tau=1}^{T(l,T)} \widehat{\mu}_{\tau-1}^l - R_\tau^l \leq Reg_{\mathcal{A}_l}(T(l,T)) + 5\sqrt{8T(l,T)\log(T^3 L/\delta)} + 1.$$

*Proof.* Let $t$ be the smallest time such that $T(l,t) = T(l,T)$. Then we have $l_t = l$, which implies that $l_t \in \mathcal{L}_t$. Therefore

$$\sum_{\tau=1}^{T(l,t-1)} \widehat{\mu}_{\tau-1}^l - \widetilde{R}_\tau^l \leq Reg_{\mathcal{A}_l}(T(l,t-1)) + 4\sqrt{8T(l,t-1)\log(T^3 L/\delta)}.$$

Further, by Lemma C.3, we have that

$$\sum_{\tau=1}^{T(l,t-1)} \widetilde{R}_\tau^l - R_\tau^l \leq \sqrt{8T(l,t-1)\log(T^3 L/\delta)}.$$

Combining with the fact that $\widehat{\mu}_{T(l,T)-1}^l - R_{T(l,t)}^l \leq 1$, we have

$$\sum_{\tau=1}^{T(l,T)} \widehat{\mu}_{\tau-1}^l - R_\tau^l \leq Reg_{\mathcal{A}_l}(T(l,T)) + 5\sqrt{8T(l,T)\log(T^3 L/\delta)} + 1.$$

This completes the proof. ∎

Now we are ready to prove Theorem 6.1.

*Proof.* The proof again condition on the 4 events listed at the beginning of the proof for Lemma C.5, which happens with at least $1 - 4\delta$ probability. We first decompose the regret as

$$\sum_{t=1}^{T} R_t^* - R_t = \sum_{l_t \neq l^*}^{T} (R_t^* - R^*) + \sum_{l_t = l^*}^{T} (R_{T(l^*,t)}^{l^*,*} - R_{T(l^*,t)}^{l^*}) + \sum_{l_t \neq l^*}^{T} (R_* - R_{T(l_t,t)}^{l_t}).$$

For the first term, we have

$$\sum_{l_t \neq l^*}^{T} (R_t^* - R^*) \leq \sqrt{8T\log(T^3 L)}.$$

For the summation for $l_t = l^*$, with Lemma C.5, we have

$$\sum_{\tau=1}^{T(l^*,T)} R_\tau^{l^*,*} - R_\tau^{l^*} \leq Reg_{\mathcal{A}_{l^*}}(T(l^*,T)).$$

For $l_t \neq l^*$, we must have $U(l_t,t) \geq U(l^*,t)$ since $l^* \in \mathcal{L}_t$ for all $t$. Then we have

$$R_* = R_* - U(l^*,t) + U(l^*,t) - U(l_t,t) + U(l_t,t)$$
$$\leq \min\left(1, \frac{Reg_{\mathcal{A}_{l^*}}(T(l^*,t-1))}{T(l^*,t-1)}\right) + \kappa\epsilon_{l^*} + U(l_t,t).$$

This follows from the second equation in Lemma C.5 and the construction of $U(l^*,t)$. Thus, we have

$$\sum_{l_t \neq l^*} R_* - R_{T(l_t,t)}^{l_t} \leq \sum_{l_t \neq l^*} \min\left(1, \frac{Reg_{\mathcal{A}_{l^*}}(T(l^*,t-1))}{T(l^*,t-1)}\right) + \kappa\epsilon_{l^*} \sum_{l \neq l^*} T(l,T)$$
$$+ \sum_{l_t \neq l^*} \left(U(l_t,t) - R_{T(l_t,t)}^{l_t}\right)$$
$$\leq Reg_{\mathcal{A}_{l^*}}(T)\log^2 T + \kappa\epsilon_{l^*}T + \sum_{l_t \neq l^*} \left(U(l_t,t) - R_{T(l_t,t)}^{l_t}\right).$$

For the term $\sum_{l_t \neq l^*} U(l_t, t) - R^{l_t}_{T(l_t, t)}$, first notice that

$$\sum_{t|l_t=l} U(l, t) - \widehat{\mu}^l_{T(l,t-1)} = \sum_{\tau=1}^{T(l,T)} \min\left(1, 2\sqrt{\frac{8\log(T^3 L/\delta)}{\tau - 1}}\right) - \kappa\epsilon_l$$
$$\leq 4\sqrt{8T(l,T)\log(T^3 L/\delta)} - \kappa\epsilon_l T(l,T).$$

Therefore, we have

$$\sum_{l_t \neq l^*} U(l_t, t) - R^{l_t}_{T(l_t, t)} = \sum_{l_t \neq l^*} U(l_t, t) - \widehat{\mu}^{l_t}_{T(l_t, t-1)} + \widehat{\mu}^{l_t}_{T(l_t, t-1)} - R^{l_t}_{T(l_t, t)}$$

$$= \sum_{l \neq l^*} \left[\sum_{t|l_t=l} U(l, t) - \widehat{\mu}^l_{T(l, t-1)} + \widehat{\mu}^l_{T(l, t-1)} - R^{i_t}_{T(i_t, t)}\right]$$

$$\leq \sum_{l \neq l^*} \left[4\sqrt{8T(l,T)\log(T^3 L/\delta)} - \kappa\epsilon_l T(l,T) + \sum_{\tau=1}^{T(l,T)} \widehat{\mu}^l_{\tau-1} - R^l_\tau\right]$$

$$\leq \sum_{l \neq l^*} \left[9\sqrt{8T(l,T)\log(T^3 L/\delta)} + Reg_{\mathcal{A}_l}(T(l,T)) - \kappa\epsilon_l T(l,T) + 1\right].$$

The last step uses Lemma C.6. Notice that

$$Reg_{\mathcal{A}_l}(T(l,T)) - \kappa\epsilon_l T(l,T) = Reg_0(T(l,T)), \quad \forall l \in [L].$$

Notice that $l \in [\frac{\log T}{2}]$. Putting everything together, we have

$$R(T) \leq \sqrt{8T\log(T^3 L/\delta)} + Reg_{\mathcal{A}_{l^*}}(T(l^*, T)) + Reg_{\mathcal{A}_{l^*}}(T)\log^2 T + \kappa\epsilon_{l^*} T$$
$$+ \frac{\log T}{2} Reg_0(T) + 5\log(T)\sqrt{8T\log(T^3 L/\delta)} + \frac{\log T}{2}$$
$$\leq \left(1 + \frac{\log T}{2} + \log^2 T\right) Reg_{\mathcal{A}_{l^*}}(T) + (1 + 5\log T)\sqrt{8T\log(T^3 L/\delta)}.$$

This completes the proof. ∎

