# OpenReview forum: "Contextual Set Selection Under Human Feedback With Model Misspecification"
_ICML.cc/2023/Workshop/ILHF — ILHF Workshop ICML 2023_

### Official Review · Reviewer_xgGW · 2023-06-03

**Rating:** 7
**Confidence:** 3

**Review:**

The paper discusses the challenge of model misspecification in stochastic contextual combinatorial  bandits, a common framework for online learning. The authors argue that the assumptions often made in learning algorithms, such as linearity, often fail to capture the complexities of real-world data, especially when human feedback is involved. To address this, they propose a robust algorithm that performs well under model misspecification. This algorithm doesn't require complete knowledge about the level of misspecification. They demonstrate, both theoretically and practically, that their algorithm avoids typical failures in the face of misspecification and outperforms a commonly used algorithm in a real-world task.


The topic of the paper is highly relevant to the workshop, and the paper has solid theoretical contributions under the realistic setup of model misspecification. The paper also shows the effectiveness of the proposed algorithms on synthetic and semi-synthetic environments. So, I think the paper should be accepted. However, the misspecification considered in the experiments are still synthetic, and in the real-world situations, we may see more unexpected type of misspecification. Therefore, it would be useful to perform more realistic experiments beyond the semi-synthetic experiments with synthesized reward functions as in the current version of the paper. It would also be interesting to extend the setting to non-linear reward function beyond the linear reward function setup as the paper does consider misspecification, but it is still based on a linear model in general.

---

### Official Review · Reviewer_FGXs · 2023-06-10
**Review of submission 14**

**Rating:** 7
**Confidence:** 3

**Review:**

This paper studies a novel setting of misspecification for combinatorial contextual bandits. Overall, this paper was well written and clearly walks through the set of proposed contributions which include a hardness result, proposed algorithm to handle misspecified setting, and then experimental validation of the proposed experiment on a synthetic dataset and movie recommendation dataset. Though I did not check the proofs in the Appendix, I found the exposition in main text on the hardness result and algorithm to be intuitive and easy to follow.

However, I had a few questions on the synthetic experiments, particularly on the movie recommendation dataset:
- Why were 30 movies recommended to users at each time step? In the combinatorial setting, shouldn't you recommend up to 30 movies?
- Can you clarify why misspecification is instantiated as a change in $\theta$, rather than via a noise term? This seems to deviate from the original problem formulation. It also seems unrealistic to that a user's reward model would completely change at a certain time step. It may be better modeled as a gradual change.

---

### Decision · Program_Chairs · 2023-06-20

Accept